

# Exploring interactions between *Beauveria* and *Metarhizium* strains through co-inoculation and responses of perennial ryegrass in a one-year trial

Milena Vera[1], Sarah Zuern[1], Carlos Henríquez-Valencia[1], Carlos Loncoman[1], Javier Canales[1,2], Frank Waller[3], Esteban Basoalto[4] and Sigisfredo Garnica[1]

[1] Instituto de Bioquímica y Microbiología, Facultad de Ciencias, Universidad Austral de Chile, Valdivia, Chile
[2] ANID–Millennium Science Initiative Program, Millennium Institute for Integrative Biology (iBio), Santiago, Chile
[3] Pharmaceutical Biology, Julius-von-Sachs Institute for Biosciences, Julius-Maximilians Universität Würzburg, Würzburg, Germany
[4] Instituto de Producción y Sanidad Vegetal, Facultad de Ciencias Agrarias y Alimentarias, Universidad Austral de Chile, Valdivia, Chile

Corresponding author
Sarah Zuern, sara.zuern@uach.cl

## ABSTRACT

Perennial ryegrass (*Lolium perenne* L.) possesses a high level of nutritional quality and is widely used as a forage species to establish permanent pastures in southern Chile. However, the productivity of most such pastures is limited by various environmental agents, such as insect pests and drought. In this context, our work stresses the need for elucidating the ability of fungal endophytes to establish interactions with plants, and to understand how these processes contribute to plant performance and fitness. Therefore, we evaluated the colonization and impact of two native strains of the endophytic insect-pathogenic fungus (EIPF) group isolated from permanent ryegrass pastures in southern Chile. Roots and seeds of ryegrass and scarabaeid larvae were collected from nine different ryegrass pastures in the Los Ríos region of southern Chile to specifically isolate EIPFs belonging to the genera *Beauveria* and *Metarhizium*. Fungal isolations were made on 2% water agar with antibiotics, and strains were identified by analyzing the entire internal transcribed spacer (ITS) 1-5.8S-ITS2 ribosomal DNA region. Four strains of *Beauveria* and 33 strains of *Metarhizium* were isolated only in scarabaeid larvae from ryegrass pastures across four sites. Experimental mini-pastures that were either not inoculated (control) or co-inoculated with conidia of the strains *Beauveria vermiconia* NRRL B-67993 (P55_1) and *Metarhizium* aff. *lepidiotae* NRRL B-67994 (M25_2) under two soil humidity levels were used. Ryegrass plants were randomly collected from the mini-pastures to characterize EIPF colonization in the roots by real-time PCR and fluorescence microscopy. Aboveground biomass was measured to analyze the putative impact of colonization on the mini-pastures' aboveground phenotypic traits with R software using a linear mixed-effects model and the ANOVA statistical test. Seasonal variation in the relative abundance of EIPFs was observed, which was similar between both strains from autumn to spring, but different in summer. In summer, the relative abundance of both EIPFs decreased under normal moisture

conditions, but it did not differ significantly under water stress. The aboveground biomass of ryegrass also increased from autumn to spring and decreased in summer in both the inoculated and control mini-pastures. Although differences were observed between moisture levels, they were not significant between the control and inoculated mini-pastures, except in July (fresh weight and leaf area) and October (dry weight). Our findings indicate that native strains of *B. vermiconia* NRRL B-67993 (P55_1) and *M.* aff. *lepidiotae* NRRL B-67994 (M25_2) colonize and co-exist in the roots of ryegrass, and these had little or no effect on the mini-pastures' aboveground biomass; however, they could have other functions, such as protection against root herbivory by insect pests.

## INTRODUCTION

Fungal endophytes are a phylogenetically diverse and widely distributed group of microorganisms that live in plant cells (hosts) as part of their life cycle, but cause no disease (*Rodriguez et al., 2009*). Some fungal endophytes participate in a highly specialized tripartite interaction with plants and pathogenic insects, which is known as endophytic insect-pathogenic fungi (EIPFs) (*Behie et al., 2017*; *Hu & Bidochka, 2019*). Members of this fungal group can act simultaneously as plant symbionts and insect pathogens (*Moonjely, Barelli & Bidochka, 2016*) and can act as natural controllers of plant insect pests. Some studies have increased our understanding of their diversity (*Spatafora et al., 2007*; *Zhang et al., 2018*), their mechanisms of infecting plant and insect hosts (*Wang & St. Leger, 2006*, *2007*) and the translocation of insect-derived nitrogen to host plants (*Behie, Zelisko & Bidochka, 2012*; *Behie & Bidochka, 2014*; *Behie et al., 2017*). However, there is a gap in the context of how EIPFs interact with plants, including aspects of colonization, abundance and impact on biomass.

Within EIPFs, the genera *Beauveria* and *Metarhizium* are among the most diverse, with global distributions and displaying significant genetic diversity, with wide insect and plant host ranges. Some differences in the below- and aboveground occurrence of these genera in an experimental cropping system (*Meyling, Thorup-Kristensen & Eilenberg, 2011*) and in different plant tissues (*Behie, Jones & Bidochka, 2015*) have been found. Thus, *Beauveria* infects aboveground insects, whereas *Metarhizium* is more commonly found infecting arthropods belowground. This pattern of natural occurrence coincides with the preferential localization of members of these genera within plant tissues. Ecologically, *Beauveria* and *Metarhizium* are well known for decreasing the plant damage caused by various phytopathogens (*Jaber & Ownley, 2018*; *Vega, 2018*) and promoting plant growth under drought conditions (*Ferus, Barta & Konôpková, 2019*; *Kuzhuppillymyal-Prabhakarankutty et al., 2020*). As reviewed in *Greenfield et al. (2016)*, species within the genera *Beauveria* and *Metarhizium* occur naturally with plants or have been artificially inoculated as endophytes in other plants. Therefore, considering the positive effects of

these fungi is important for expanding their role in sustainable agriculture (*Branine, Bazzicalupo & Branco, 2019*). Perennial ryegrass (*Lolium perenne* L.) is a temperate grass species commonly used as pasture for livestock in southern Chile, which is affected by endemic insect pests (*Cisternas, 1989*) and by drought during the summer (*Iglesias, 2012*). Although *Beauveria* and *Metarhizium* have been detected in soils planted with perennial ryegrass (*Kolczarek, 2015*), to our knowledge, no study has explored their interactions with this host plant, changes in colonization abundance in the roots over time and the impact of entomopathogenic fungi on ryegrass growing under water restriction conditions comprising at least one annual production cycle.

In this study, plants and scarabaeid larvae were collected from nine perennial ryegrass pastures in the Los Rios region (Chile) allowing us to detect root-, seed- and insect-associated fungi, where a substantial fraction of EIPF isolates were gathered. We were interested about the nature of the interactions between EIPF strains and ryegrass plants. Therefore, this study focused on interactions induced through the co-inoculation of two native fungal EIPF strains belonging to the genera *Beauveria* and *Metarhizium* on perennial ryegrass growing in mini-pastures with natural soil and under normal water availability and drought stress. Using these mini-pastures, we addressed the following specific questions: (i) Is there a difference between the EIPF strains in their ability to colonize ryegrass roots? (ii) Is there an increase in the relative abundance of the EIPF strains in fine roots over time? (iii) Does co-inoculation with EIPF strains have an impact on the aboveground biomass of perennial ryegrass mini-pastures?

## MATERIALS AND METHODS

### Collection of plants and insects

Plants and seeds of perennial ryegrass (*Lolium perenne* L.), and living and dead (Coleoptera: Scarabaeidae) larvae were collected during November–December 2018 (roots and larvae) and March–April 2019 (roots and seeds) from nine pasture sites near roads or railways in the Los Rios region, Chile (Figs. 1A–1E). The sampling design included three latitudinal transects from the Andes to the Coastal Cordillera: one transect in the south (a–c), one transect in the middle (d–f) and one transect in the northern part (g–i) of the region. Through this strategy, we attempted to sample sites from different soil series and grazing pasture managements. Thus, 180 plant individuals were collected in total (3 transects × 3 plots × 10 plant individuals × 2 collection periods), as well as 14 dead and 164 living white grubs corresponding to the larvae of scarabaeids that feed on organic matter and the roots of ryegrass. For the sites h, f and d we conducted this research under sampling permits from Juan Carlos Marin, Luis Toloza and Carlos Villagra, respectively.

### Fungal isolation from plants and larvae at field sites

Ryegrass seeds and roots were first surface washed with running water to remove material adhering to the surface. Subsequently, they were immersed in 1% sodium hypochlorite (NaClO) for 5 min and then thoroughly washed five times with sterile distilled water. Root fragments and seeds as well as fungal structures growing on field-dead larvae were aseptically placed onto 90 mm × 14.5 mm Petri dishes containing 2% water agar
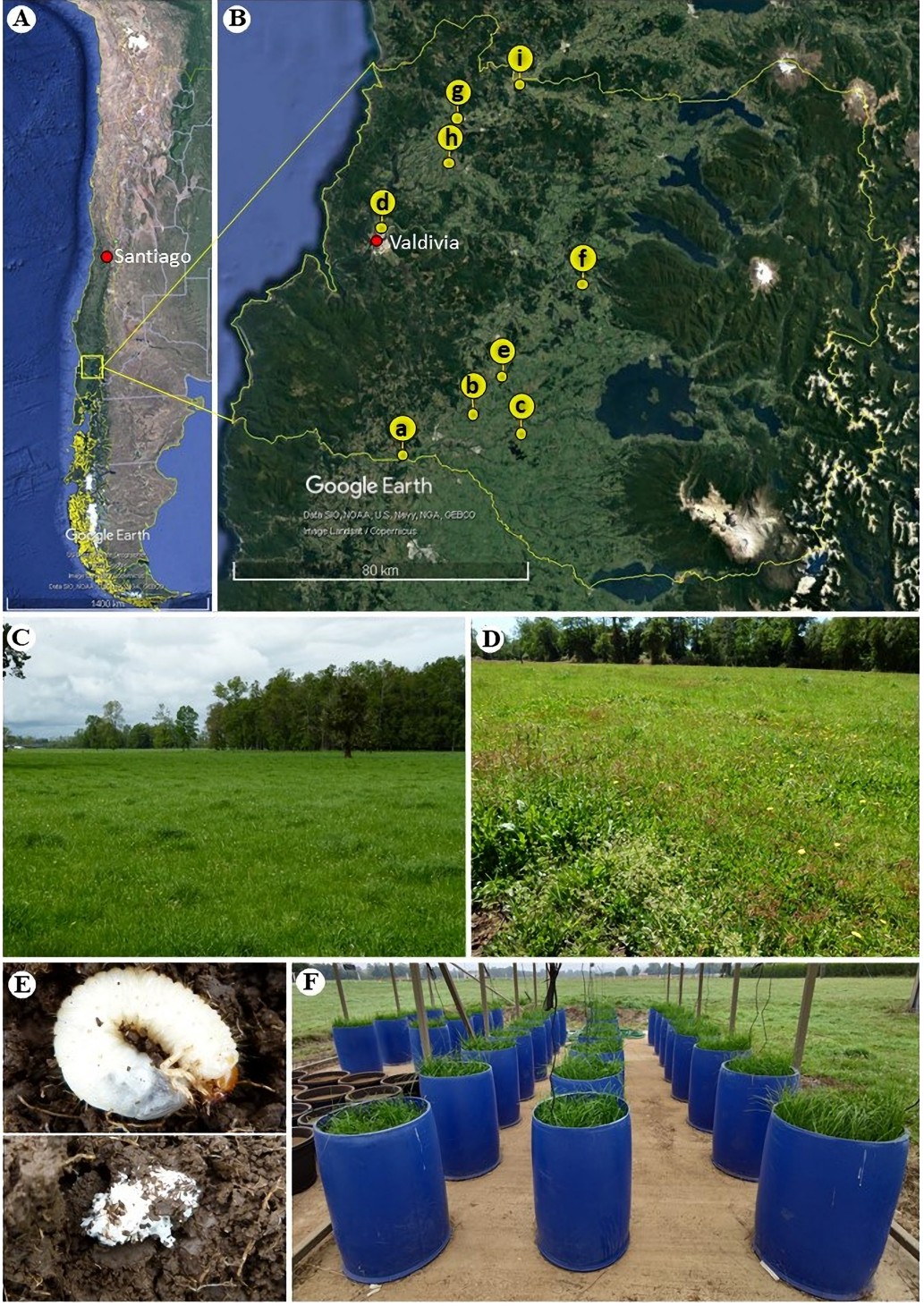

**Figure 1 Geographic location and sampling sites of ryegrass (*Lolium perenne*) and larvae of scarabaeids in the Los Ríos region (Chile).** (A) Map of Chile, indicating the region where the material was collected (map data: Google Earth, ©2021). (B) Collection sites across the Los Ríos region: (a) Road: T-85 La Union to Alerce Costero, (b) Road: T-706 sector Rapaco; (c) Road: T-85 Rio Bueno to Lago Ranco; (d) Santa Rosa experimental station; (e) Paillaco-Rucaquilen; (f) Pichinhue, (g) Road: T-20

**Figure 1** (continued)
San José de la Mariquina to Mehuin, (h) pasture in Mafil; (i) site near a railway line in Lanco (map data: Google Earth, ©2021). (C) Pasture with ryegrass lacking larvae predation. (D) Ryegrass pasture effected by larvae predation. (E) Live scarabaeid larvae in the soil (above) and dead scarabaeid larvae in the soil at the time of collection (below). (F) Set-up of the mini-pasture experiment with humidity sensors.

supplemented with antibiotics (0.4 g/200 mL penicillin G sodium salt and 0.9 mg/200 mL streptomycin sulfate) and incubated at room temperature. Living larvae were cleaned superficially in a 1% sodium hypochlorite solution and rinsed five times with sterile distilled water and subsequently killed by thermic shock and maintained in a humid chamber for 14 days to allow the growth of entomopathogenic fungi.

Fungal growth was checked periodically by observation with a stereo microscope. Once mycelia appeared either from seeds, roots and/or larvae, they were transferred to Petri dishes containing potato dextrose agar (PDA) (Merck KGaA, Darmstadt, Germany) and the cultures were incubated at room temperature until the colonies reached a diameter of 20 to 30 mm. Fungal strains were identified on the basis of their colony, morphological and reproductive characteristics after subculturing on potato dextrose agar (PDA) plates (*von Arx, 1982*). Isolates morphologically identified as *Beauveria* and *Metarhizium* were subsequently sequenced to obtain a more accurate identification.

## DNA extraction, PCR and sequencing

The mycelium was removed from the surface of fresh cultures grown in PDA and deposited in a 2-mL microcentrifuge tube and pulverized in liquid nitrogen with a pestle. To remove external fungal infections on ryegrass seeds, 1 g of seeds was sterilized as described above and pulverized in liquid nitrogen by grinding with a mortar and pestle. The total genomic DNA was obtained with an E.Z.N.A. Fungal DNA Mini Kit (Omega Bio-tek, Norcross, GA, USA), according to the manufacturer's instructions. To amplify the nuclear segment including the ITS1–5.8S–ITS2 region (ITS), we used the primer combination ITS1F (*Gardes & Bruns, 1993*) and ITS4 (*White et al., 1990*). We used a touchdown PCR with an initial step of 94 °C for 3 min, followed by 10 cycles of 94 °C for 30 s, with the annealing temperatures starting at 60 °C for 45 s (decreasing by 1 °C per cycle), then 72 °C for 1 min 15 s for elongation, followed by 26 cycles of 94 °C for 30 s, 50 °C for 45 s, elongation at 72 °C for 1 min 15 s and a final extension at 72 °C for 7 min. The same primers used for the PCR were used to sequence both strands of DNA in an automated genetic analyser (ABI3500; Applied Biosystems, Foster City, CA, USA) of the AUSTRAL-*omics* core facility at the Universidad Austral de Chile (www.australomics.cl, accessed 1 July 2021). Sequencher 4.1 (Gene Codes Corporation, Ann Arbor, MI, USA) was used to assemble and edit the sequence data.

## Antagonism assays, BLAST, sequence alignments and phylogenetic analyses

To detect whether there was an inhibitory effect between the isolated EIPF strains (Table S1), a dual-culture antagonism tests on 2% PDA. For initial identification, ITS

sequences from all these strains were compared against sequences from GenBank by BLAST (*Altschul et al., 1990*). To carry out the experimental assays, we selected the strains *Beauveria vermiconia* NRRL B-67993 (P55_1) and *Metarhizium* aff. *lepidiotae* NRRL B-67994 (M25_2), which have been deposited at the NRRL Culture Collection (https:// nrrl.ncaur.usda.gov/, accessed 1 July 2021). To infer the phylogenetics of the fungal strains, we assembled an alignment (Dataset 1 for the genus *Beauveria* and Dataset 2 for the genus *Metarhizium*) including all available sequences from both genera and the corresponding outgroups in GenBank (https://www.ncbi.nlm.nih.gov/, accessed 1 July 2021).

Both datasets were aligned separately with MAFFT v7.407 under the E-INS-i option (*Katoh & Standley, 2013*). Subsequently, the optimal models for each dataset were identified with PhyML 3.0 (*Guindon et al., 2010*) available at the ATGC: Montpellier Bioinformatics platform (http://www.atgc-montpellier.fr, accessed 1 July 2021). In addition, maximum likelihood phylogenetic trees were inferred with RAxML 7.0.3 (*Stamatakis, 2006*) and the GTR+I+G model of nucleotide substitution for both datasets. Tree searches were carried out with 1,000 rounds generated from distinct maximum parsimony starting trees. Bootstrap support for individual branches was assessed with 1,000 replicates (*Felsenstein, 1985*). Phylogenetic trees were viewed and rooted with FigTree v1.4.4 (*Rambaut, 2009*).

## Experimental design and harvest

Seeds from nine cultivars of perennial ryegrass commercialized in Chile were tested for germination and microbial load (fungal and bacterial contamination) (Table S2). The seeds were sterilized as described above and 100 seeds per perennial ryegrass cultivar were placed in Petri dishes containing water agar (SIGMA, St. Louis, MO, USA) at 2% and incubated for 14 days at ~23 °C. Observations of seed germination and microbial contamination were performed at 7 and 14 days after inoculation. According to these assays, the cultivar RODEO Edge was selected for this study as it had the highest seed germination percentage and the lowest presence of microorganisms.

Twenty-four perennial ryegrass mini-pastures were established in 120-L containers 40 cm in diameter at the beginning of May 2020 (Fig. 1F). The experiments were mounted in an open-walled greenhouse with a transparent plastic roof; this way, the temperature and environmental humidity corresponded to those of the local climatic conditions (average annual temperature, 12 °C; rainfall up to 2,500 mm, mostly in the winter (see *Huber (1970)*) present in the EEAA belonging to the Universidad Austral de Chile located in sector Cabo Blanco, 4 km north of Valdivia (Chile). The soil comes from a naturalized grassland vegetation and corresponds to the Andisol type (Typic Hapludand) belonging the Valdivia series with a high organic matter content, a low pH and high phosphorus retention (*CIREN, 2003*). Before the experiments started, a chemical analysis of the soil was carried out at the Soil Laboratory of the Universidad Austral de Chile (Table S3). Fertilization was carried out 70 days after planting, in which N, P and K were supplied by a commercial granulated mixture of urea (1 g), triple superphosphate (15 g) and Basacote Green Tec Truf K (13.9 g) in each mini-pasture. After 205 days, N

continued to be supplied to each mini-pasture as urea (0.8 g), once after each cutting, until experiment finished.

In each pot, 3.5 g of seeds that had been sterilized as described above were distributed evenly across the surface and covered with a thin layer of soil. Twelve randomly selected pots were co-inoculated with the EIPF strains; the remaining 12 pots corresponded to the control treatment (lacking EIFP strains). For fungal inoculation, strains were routinely grown in 500-mL Erlenmeyer flasks containing 200 g of sterilized rice at 23 °C for 14 days. Conidial suspensions were prepared by adding 250 mL of sterile distilled water plus 0.1% Tween 80 to the Erlenmeyer flasks. The conidial suspensions of each fungal strain were then homogenized and filtered into a plastic bottle through a parafilm with very small perforations. The conidia concentration in each fungal suspension was determined under a light microscope in a Neubauer chamber and adjusted to a concentration of $1 \times 10^8$ conidia/mL (modified from *Jaber & Enkerli, 2016*). Using the soil drench technique, the surface of the pots was drenched with 250 mL of a previously prepared 1:1 conidia suspension of each strain (modified from *Greenfield et al., 2016*). During the growing season, the mini-pastures were regularly watered with 2,000–3,000 mL of tap water per pot to reach 80–85% field capacity (FC), representing water availability conditions. At the beginning of summer, a water restriction scenario representing the summer season was imposed on six mini-pastures with and six without inoculation with EIPF. The limit of irrigation was 40–45% FC. At the start of the water restriction treatment, the soil moisture level was monitored by TEROS 11 moisture sensors (METER Group; Pullman Inc., Chicago, IL, USA) installed at 20 cm depth.

Consecutive harvests were carried out by hand when the growing degree-days reached 270 degrees (to obtain plants with three leaves). Plants were cut to 4 cm high to ensure good regrowth in the mini-pastures. In the aboveground biomass obtained through harvesting, fresh weight and leaf area index measurements were made. The harvested material was deposited in ovens and left to dry for 3 days at 60 °C to determine the dry weight. In order to carry out microscopic and molecular analyses, six plants from each mini-pasture were randomly sampled every 2 months after the start of the experiment in May (autumn), corresponding to June (autumn), August (winter) and October (spring) (normal water availability), and 1 month under drought stress in January and March (summer). In each sample, six plants from each treatment were carefully collected at random, deposited in labelled hermetic plastic bags and stored in the laboratory at –20 °C for further analysis.

## Quantification of root colonization by EIPF

EIPF-inoculated and non-inoculated perennial ryegrass plants were collected from the mini-pastures and placed separately into plastic bags to be transported to the laboratory. The roots were separated and thoroughly washed with distilled water until all the soil particles adhering to the roots were removed. The fresh weight and dry weight in fine roots ranged from 0.02 to 0.3 g on average within the same sample. Once dried, they were weighed and placed in 2-mL Eppendorf tubes. Subsequently, the roots samples were pulverized in liquid nitrogen with the help of a pestle and the genomic DNA was extracted

**Table 1 Primers used for quantitative PCR detection of *Beauveria vermiconia* NRRL B-67993 and *Metarhizium* aff. *lepidiotae* NRRL B-67994 colonization in ryegrass roots.** Target regions are *B. vermiconia* spacer 2, *M. aff. lepidiotae* spacer 1 and *L. perenne* spacer 2.

| Species | Primer name | Primer sequence (5′–3′) | Length (bp) | Tm (°C) | CG% | Product length (bp) |
|---|---|---|---|---|---|---|
| *B. vermiconia* | BEA | (F) CCTAGGAAGTCGGCATTGGG | 20 | 60.18 | 60 | 138 |
| | | (R) AAGTTGGGTGTTTTACGGCG | 20 | 59.34 | 50 | |
| *M. aff lepidiotae* | MET | (F) CCTGTTCGAGCGTCATTACG | 20 | 59.09 | 55 | 150 |
| | | (R) TCCTGTTGCGAGTGTTTTAC | 20 | 56.3 | 45 | |
| *L. perenne* | LOL | (F) GGCGGCATCGTCCGTCGCTT | 20 | 68.19 | 70 | 202 |
| | | (R) GAGAGCCGAGATATCCGTTGCC | 22 | 62.84 | 59 | |

with the DNAeasy PowerSoil Pro Kit (QIAGEN, Hilden, Germany). The extracted genomic DNA was stored at –20 °C for later use.

The relative amount of fungal DNA in these samples was determined by performing quantitative PCR reactions with a *B. vermiconia* NRRL B-67993-specific primer pair, a *M.* aff. *lepidiotae* NRRL B-67994-specific primer pair and a perennial ryegrass-specific primer pair. Subsequently, the efficiency-corrected Ct values for each fungal strain were subtracted from those of the perennial ryegrass. The efficiency correction is required for accurate quantitative PCR analysis, as recently suggested by *Ruijter et al. (2021)*. PCR primers were designed with Primer-BLAST (*Ye et al., 2012*) (Table 1), in which the *B. vermiconia* NRRL B-67993-specific primer pair (*BEA*) binds to ITS2, the *M.* aff. *lepidiotae* NRRL B-67994-specific primer pair (*MET*) binds to ITS1 and the *LOL* primer pair binds to the ITS2 of perennial ryegrass.

For PCR amplification, ThermoFisher SYBR Green Power Up (ABGene; www.thermofisher.com) was used in a final volume of 20 μL, following the manufacturer's instructions. Real-time PCR was performed in a QuantStudio 3 Thermocycler (Applied Biosystems, Foster City, CA, USA). The qPCR assays were performed using reactions in triplicate in a total volume of 20 μL containing 10 μL of ThermoFisher SYBR Green Power Up, 0.4 μL of the forward and reverse primers at 0.2 μM, 7.2 μL of nuclease-free water and 2 μL of genomic DNA. The parameters for qPCR were a first step of activation at 50 °C for 2 min, 2 min at 95 °C and 40 cycles of denaturation at 95 °C for 15 s, and alignment and extension at 60 °C for 30 s. To estimate the amount of DNA in the MET and BEA target fragments within the root samples and its variation over time (Samples 1 to 5), relative DNA quantification was carried out. In the first instance, an attempt was made to use the relative quantification of the fungi that assumed an amplification efficiency between 90% and 110%, then compared the exponential curves of ΔCt ($2^{\Delta Ct}$) directly (*Livak & Schmittgen, 2001*). However, because it is an environmental material, we had difficulty achieving the optimal values required for all primer pairs and samples. For this reason, we used the Miner software (*Zhao & Fernald, 2005*) to obtain the efficiency and Ct values for qPCR from individual PCR reactions. These values were used to calculate the relative abundance ($R\_0$) of the target genes (*MET* and *BEA*) normalized to the plant gene (*LOL*) considering the average efficiency for each primer pair.

For microscopic examination of fungal root colonization, fine roots of perennial ryegrass were rinsed in water as described above, and fixed in a 1:3 mixture (vol/vol) of Chloroform and Ethanol containing Trichloroacetic acid (1.5 g/L). Subsequently, roots were washed with phosphate buffer and stained with Wheat Germ Agglutinin-Alexafluor 488 (Molecular Probes; Invitrogen, Waltham, MA, USA) as described in *Deshmukh et al. (2006)*. Images were obtained with a Leica TCS SP5 confocal microscope using the brightfield channel and a Green Fluorescent Protein (GFP) filter set (excitation wavelength 488 nm, bandpass 505–530 nm) for the detection of Wheat Germ Agglutinin-Alexafluor 488. Final images were prepared as maximum projections of z-stacks, if not stated otherwise.

### Data analysis

First, we checked whether the fungal inoculation procedure resulted in higher colonization rates and if there were differences in the abundance between the two inoculated fungal strains. We used the relative amount of fungi in the roots, as determined by qPCR performed with DNA from the root samples.

The variability of the relative abundance of DNA was analyzed by means of a linear mixed model (following a normal distribution of the data), using the variables time (month of sampling), fungal strain (*M*. aff. *lepidiotae* and *B. vermiconia*) and water treatment (80–85% FC and 40–45% FC) as fixed variables, whereas the factor "mini-pasture" was incorporated as a random variable. In the same way, a linear mixed model was used to explain the variability in the biomass production of the mini-pastures, according to the fresh weight, dry weight and leaf area. The fixed variables for the biomass model were time, fungal inoculation (inoculated mini-pastures *vs*. control mini-pastures) and water treatment. The "mini-pasture" factor was incorporated as a random variable. ANOVA was used to determine the statistical significance of the data, with *P*-values less than 0.05 considered significant. Tukey's test was used to identify the levels between which significant differences were obtained for the variable "time".

Linear mixed models for each response variable were constructed by using the lmer function of the lme4 data package (*Bates et al., 2015*) and validated according to the Akaike criterion (AIC). Both linear mixed models and ANOVA were performed in R Studio (*R Core Team, 2018*). Graphical representation of the data was carried out in GraphPad Prism 8.

## RESULTS

### EIPF isolates, antagonism assay and phylogenetic identification

Four *Beauveria* and 33 *Metarhizium* strains were isolated from insect larvae collected from Sites b, d, e, h and i (see Fig. 1B and Table S4). No *Beauveria* and *Metarhizium* isolates were obtained from roots and seeds of perennial ryegrass. From each sample, one strain was selected for antagonism testing, resulting in two *Beauveria* and 19 *Metarhizium* strains. All dual cultures that formed more than one colony per plate were discarded from the measurements. A similar growth pattern was observed for all *Metarhizium vs. Beauveria* strains, showing a percentage inhibition between 50% and 60%. In contrast, the

*Beauveria* strains had different growth patterns: strain S01_6 showed growth differences in dual cultures compared with the control, whereas strain P55_1 had homogeneous growth in dual cultures, with a percentage inhibition of 47%. (Table S1). Finally, the combination of *B. vermiconia* NRRL B-67993 (P55_1) and *M.* aff. *lepidiotae* NRRL B-67994 (M25_2) was selected because of the lower percentage of inhibition compared with the controls.

Our ITS phylogenetic analyses showed that the strain NRRL B-67993 was 100% identical to the collection ARSEF 2922 (type material of *B. vermiconia*), whereas the strain NRRL B-67994 has an isolated position, forming a cluster with the collection ARSEF 7488 (type material of *M. lepidiotae*) supported by an 83% bootstrap value (Fig. 2).

## Primer specificity and quantitative PCR (qPCR) performance

The primers used in the qPCR assays for the relative quantification of *B. vermiconia*, *M.* aff. *lepidiotae* and perennial ryegrass are shown in Table 1. All three primer pairs were free from non-specific amplification, and the amplicons were confirmed to be of the correct size. In addition, the ITS primers were specific to perennial ryegrass and did not amplify any of the *Beauveria* and *Metarhizium* strains tested.

## EIPF colonization of perennial ryegrass roots as detected by fluorescence microscopy and real-time PCR

Real-time PCR assays showed amplification of fungal DNA in inoculated plants (Table S5), and no fungal DNA was detected in non-inoculated control plants.

The linear mixed model indicated the significant effect of the variable time (months) on the colonization of ryegrass roots by EIPFs (Table 2). Figure 3A shows the relative abundance of *B. vermiconia* NRRL b-67993 (P55_1) and *M.* aff. *lepidiotae* NRRL b-67994 (M25_2) prior to the incorporation of the water restriction treatment (soil moisture at 40–45% FC), corresponding to the months of June (autumn), August (winter) and October (spring). During this period, all the mini-pastures were maintained at the same soil moisture (80–85% FC). From June to August, a significant decrease in the abundance of both fungi was observed (Tukey's test, $P = 0.001$), but from August to October, the abundance increased significantly (Tukey's test, $P = 0.007$). No significant differences were observed when we compared the abundance of both fungi in any of these 3 months. In December (early summer), the mini-pastures were divided into two blocks. One block was maintained with a water supply corresponding to a soil moisture content of 80–85% FC, and the other block was supplied with less water to reach a soil moisture content of 40–45% FC. According to the linear mixed model, time and fungal strain were the determining variables in ryegrass root colonization in January and March, corresponding to the summer season (Table 2). Under the 80–85% FC humidity treatment, both strains significantly decreased their relative abundance from January to March (Tukey's test, $P = 0.005$ and $P = 0.011$ for *M.* aff. *lepidiotae* NRRL b-67994 (M25_2) and *B. vermiconia* NRRL b-67993 (P55_1), respectively). However, in both January and March, *M.* aff. *lepidiotae* NRRL b-67994 (M25_2) was significantly more abundant than *B. vermiconia* NRRL b-67993 (P55_1) (Tukey's test, $P = 0.044$ and $P = 0.014$, respectively). Under the

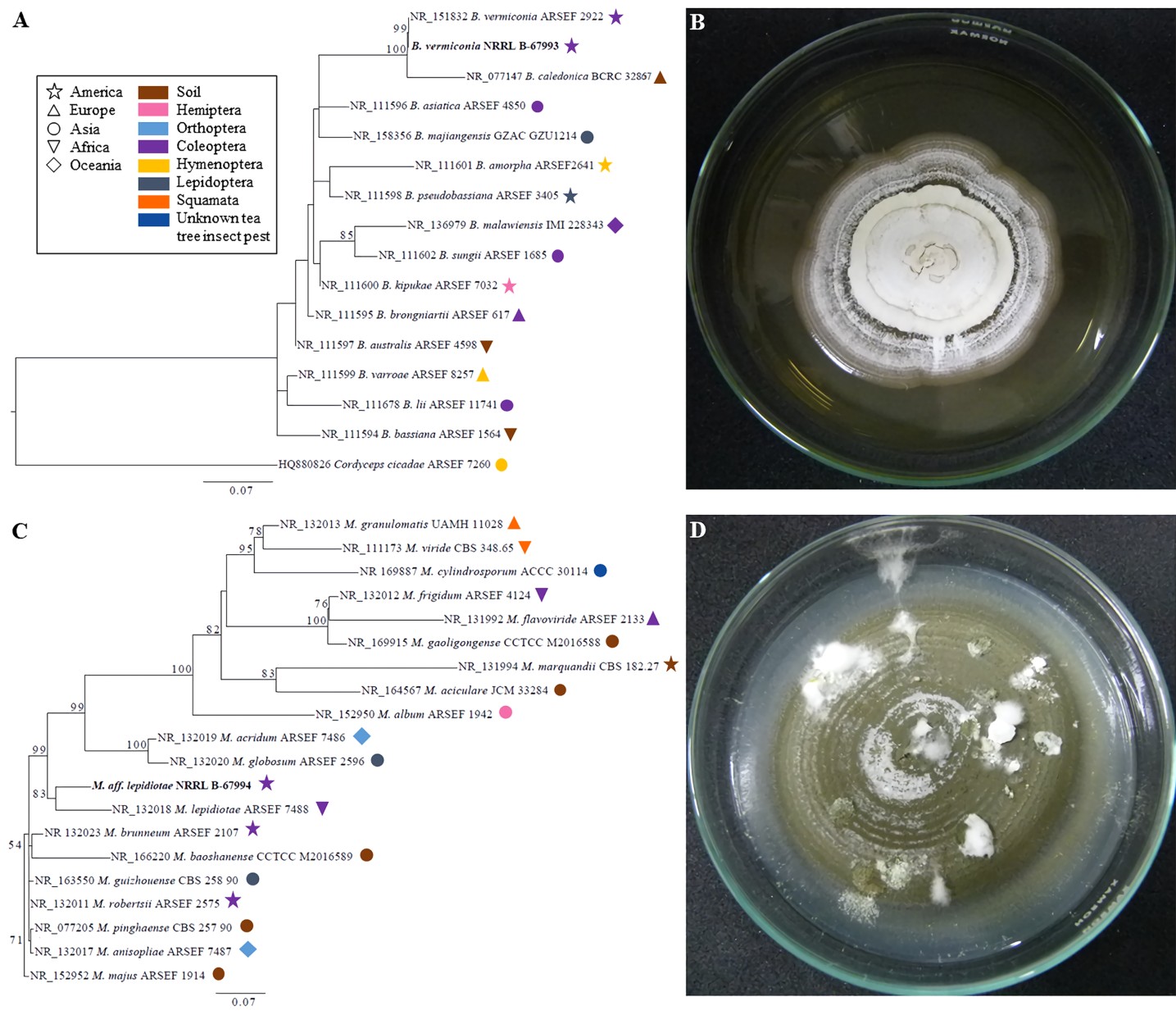

**Figure 2** **Phylogenetic placement of *Beauveria vermiconia* NRRL B-67993 and *Metarhizium* aff. *lepidiotae* NRRL B-67994, based on nuclear rDNA ITS sequences and their cultural features.** (A) The best maximum likelihood trees from the ITS sequences from the genus *Beauveria*, depicting the phylogenetic placement of the strain *B. vermiconia* NRRL B-67993 obtained from 1,000 runs under the GTR+G+I model. The species *Cordyceps cicadae* ARSEF 7260 was used out-group. Bootstrap support values ≥50 are given on the branches. For each sequence, both the substrate and continent of origin are coded on the branches. (B) *Beauveria vermiconia* NRRL B-67993 grown on a 2% PDA medium at 55 days. (C) The best maximum likelihood trees from ITS sequences from the genus *Metarhizium*, depicting the phylogenetic placement of the strain *M.* aff. *lepidiotae* NRRL B-67994 obtained from 1,000 runs under the GTR+G+I model is illustrated. The species *Metarhizium majus* ARSEF 914 was used as outgroup. Bootstrap support values ≥50 are given on the branches. For each sequence, both the substrate and continent of origin are coded on the branches. (D) *Metarhizium* aff. *lepidiotae* NRRL B-67994 grown on a 2% PDA medium at 55 days.

40–45% FC treatment, the strains showed significant differences in relation to their relative abundance only in March (Tukey's test, $P = 0.026$), with *M.* aff. *lepidiotae* NRRL b-67994 (M25_2) again being more abundant than *B. vermiconia* NRRL b-67993 (P55_1);

**Table 2 Analysis of variance testing the effects of time (months), EIPF strains (*Metarhizium* aff. *lepidiotae* and *Beauveria vermiconia*) and water restriction (80–85% FC and 40–45% FC) on the relative abundance of fungal rDNA in ryegrass roots.**

| | Relative abundance of fungal rDNA | | | Above biomass | | | | | | |
| --- | --- | --- | --- | --- | --- | --- | --- | --- | --- | --- |
| | | | | | Fresh weight | | Dry weight | | Leaf area | |
| | df | *F*-ratio | *P*-value | df | *F*-ratio | *P*-value | *F*-ratio | *P*-value | *F*-ratio | *P*-value |
| Normal water availability: | | | | | | | | | | |
| Time | 2 | 5.051 | **0.009** | 4 | 396.251 | **<0.001** | 782.907 | **<0.001** | 32.243 | **<0.001** |
| EIPF strains | 1 | 0.084 | 0.772 | 1 | | | | | | |
| Inoculation treatment | | | | | 3.342 | 0.070 | 2.170 | 0.144 | 1.022 | 0.314 |
| Time × EIPF strains | 2 | 0.619 | 0.541 | 4 | | | | | | |
| Time × Inoculation treatment | | | | | 1.902 | 0.115 | 3.890 | **0.005** | 0.388 | 0.817 |
| Water restriction: | | | | | | | | | | |
| Time | 1 | 15.5852 | **<0.001** | 2 | 33.3719 | **<0.001** | 13.7201 | **<0.001** | 2.4096 | 0.09822 |
| EIPF strains | 1 | 7.2791 | **0.01** | | | | | | | |
| Inoculation treatment | | | | 1 | 0.2866 | 0.5943 | 1.8366 | 0.18027 | 0.2612 | 0.61108 |
| Water treatment | 1 | 1.3589 | 0.25082 | 1 | 5.64 | **0.02** | 8.0237 | **0.00622** | 1.4661 | 0.23056 |
| Time × EIPF strains | 1 | 0 | 0.99772 | | | | | | | |
| Time × Inoculation treatment | | | | 2 | 0.1191 | 0.88789 | 0.3385 | 0.71417 | 0.283 | 0.75451 |
| Time × Water treatments | 1 | 3.0243 | 0.08991 | 2 | 15.6127 | **<0.001** | 0.0834 | **<0.001** | 4.1459 | **0.02042** |
| Water treatment × EIPF strains | 1 | 1.4931 | 0.22907 | | | | | | | |
| Water treatment × Inoculation treatment | | | | 1 | 0.443 | 0.5081 | 16.19 | **<0.001** | 2.1128 | 0.15112 |

**Note:**
For each linear mixed model, the variables time, EIPF strain, inoculation treatment and water restriction were considered as fixed effects and mini-pastures as random variables. Significant *P*-values are in bold (*P* < 0.05).

additionally, their abundance at this moisture level was significantly higher than under the 80–85% FC moisture treatment (*P* = 0.008) (Fig. 3B). When we compared the total fungal rDNA (rDNA *M.* aff. *lepidiotae* NRRL b-67994 (M25_2) + rDNA *B. vermiconia* NRRL b-67993 (P55_1)), a significant decrease in total abundance was observed from January to March under the 80–85% FC humidity treatment (*P* = 0.001), which, in turn, was significantly lower than that under the 40–45% FC humidity treatment (*P* = 0.005), but only in March (Fig. 3B). The colonisation of perennial ryegrass roots was confirmed by staining chitin-containing fungal cell walls with WGA-Alexafluor. In root samples from inoculated plants, hyphal structures could be observed in all analyzed roots, sometimes forming extensive networks on the root surface (Figs. 3C and 3D). Hyphae were also frequently observed to grow inter- and intracellularly in the rhizodermis (Fig. 3E and Fig. S1).

## Impact of EIPFs on aboveground perennial ryegrass biomass in mini-pastures

In contrast to the rDNA analyses, biomass measurements were made in July (winter), September (early spring), October (spring), November (spring–summer transition), December (early summer), January (summer), February (summer) and March (summer–autumn transition) (Table S6). Data from July to December (Figs. 4A, 4C and

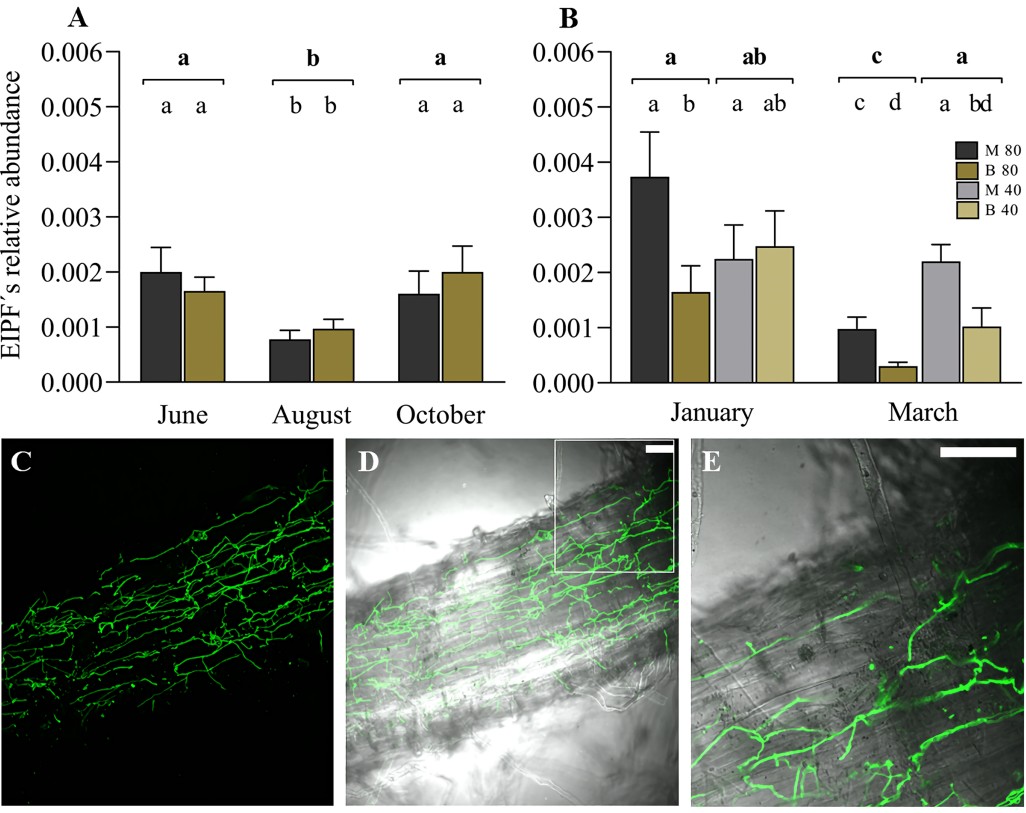

**Figure 3 Experimental co-inoculation of *ryegrass* mini-pastures with *Beauveria vermiconia* NRRL B-67993 and *Metarhizium* aff. *lepidiotae* NRRL B-67994, and root colonization.** (A) Relative abundance of endophytic insects and pathogenic fungi (rDNA EIPFs/rDNA *L. perenne*) in ryegrass mini-pastures with a soil moisture content of 80–85% field capacity (FC) estimated by a qPCR-based approach. Estimated means ($n = 12$) and standard errors (means with SEM) are presented. (B) Relative abundance of EIPFs under a water restriction treatment (80–85% FC *vs.* 40–45% FC) based on qPCR analyses. Estimated means (n=6) and standard errors (means with SEM) are presented. Bold letters above the black line indicate differences between groups and different letters above the bars indicate differences between fungal strains (95% level of confidence ($P < 0.05$)). (C–E) Confocal fluorescence images of a root section of a *Lolium perenne* plant 8 weeks after inoculation, stained with Wheat Germ Agglutinin-Alexafluor 488 (WGA-AF 488). Fluorescence image (C) and overlay of fluorescence image and bright field image (D). A magnified portion of one z-slide of image D is shown in (E). Scale bar represents 50 μm.

4E) correspond to the period of soil moisture at 80–85% FC, and data from January to March included 50% of the mini-pastures under water stress. For biomass production, the linear mixed model indicated that the variable "time" was highly significant for the annual variation in fresh weight, dry weight and leaf area, and the interaction of time and inoculation treatment was significant for dry weight (Table 2). The growth pattern of ryegrass was characterized by significant monthly increases in fresh and dry weights towards spring (July–November), which subsequently decreased towards summer (December), contrary to how the leaf area decreased from July to November and increased towards December. Specifically, Tukey's test revealed that in July, the fresh weight was significantly higher in the mini-pastures inoculated with the EIPFs ($P = 0.015$), but showed a significantly lower leaf area than the control mini-pastures ($P = 0.032$), although the dry

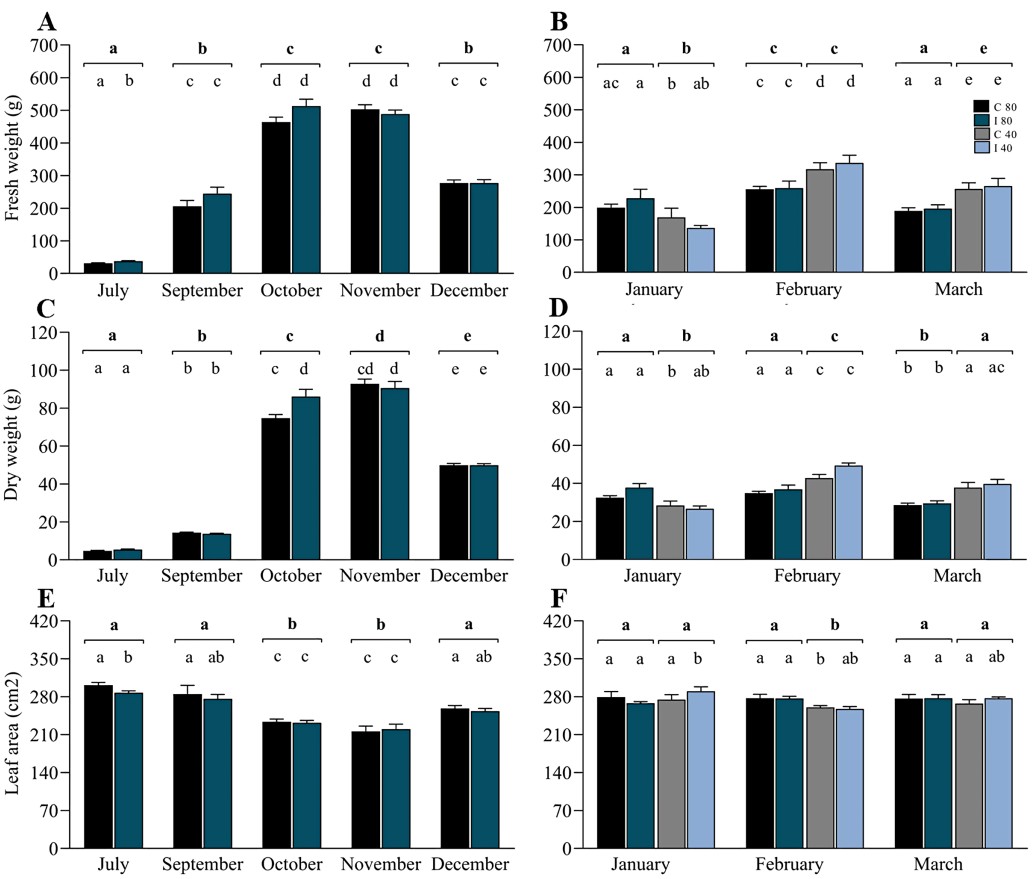

**Figure 4 Co-inoculation with *Beauveria vermiconia* NRRL B-67993 and *Metarhizium* aff. *lepidiotae* NRRL B-67994 and its impact on the aboveground biomass of ryegrass grown in mini-pastures under semi-controlled conditions.** (A), (C) and (E) correspond to the ryegrass mini-pastures growing at 80–85% field capacity (FC), whereas (B), (D) and (F) correspond to comparisons of treatments with 80–85% FC *vs.* ryegrass mini-pastures grown at 40–45% FC. Bold letters above the black line indicate differences between groups, and different letters above the bars indicate differences between control and inoculated mini-pastures (95% confidence ($P < 0.05$)). Estimated means ($n = 12$ and $n = 6$) and standard error (mean with SEM) are presented. Abbreviations are as follows; C 80 = mini-pastures control at 80–85% FC soil humidity, C 40 = mini-pastures control at 40–45% FC humidity soil, I 80 = mini-pastures inoculated at 80–85% FC soil humidity and I 40 = mini-pastures inoculated at 40–45% FC soil humidity.

weight in October was significantly lower in the control mini-pastures than in the inoculated mini-pastures ($P = 0.013$). From January to March (Figs. 4B, 4D and 4F), the linear mixed model indicates that the hydric treatment was the most explanatory for biomass variation (Table 2), since in all 3 months, significant differences were observed between moisture levels for both fresh and dry weight, and for leaf area only in February. Specifically, in January, fresh and dry weights were significantly lower in the mini-pastures with a lower water supply (Tukey's test, $P = 0.007$ for both variables). However, in February and March, mini-pastures with lower soil moisture produced more biomass than mini-pastures with higher moisture (Tukey's test, $P$-values for February and March: $P = 0.001$ and $P < 0.001$ for fresh weight, and $P < 0.001$ and $P = 0.001$ for dry weight, respectively). Leaf area was significantly lower in the water-restricted mini-pastures during

February (Tukey's test, $P < 0.001$). No significant differences were observed when we compared control *vs.* inoculated mini-pastures under the same water treatment and in the month, but significant differences were observed when we compared control mini-pastures between months for each water treatment, and when we compared inoculated mini-pastures under the same experimental conditions. In Figs. 4B, 4D and 4F, these statistically significant differences can be observed according to the letters representing the significance codes.

## DISCUSSION

Our results document that EIPF communities are relatively common in the sampled agroecosystems. These findings agree with previous studies indicating that these fungi are widely distributed and frequent in agricultural systems (*Steinwender et al., 2014*, *2015*; *Hernández-Domínguez & Guzmán-Franco, 2017*; *Iwanicki, Pereira & Botelho, 2019*). The occurrence of EIPFs has been reported in soils under perennial ryegrass cultivation (*Kolczarek, 2015*). Specifically, members of the genera *Beauveria* and *Metarhizium* were frequent on insects, which could indicate that they are directly involved in belowground interactions in such grasslands. In Chile, previous studies have detected the wide distribution of the genera *Beauveria* and *Metarhizium*, isolated from different substrates and localities (*e.g.*, *France et al., 2000*; *Becerra et al., 2007*; *Velásquez et al., 2007*). In our study, we were unable to isolate these genera from ryegrass roots and seeds; however, this does not mean that these fungi were not present on such materials. A plausible explanation for this result could be that other, faster-growing fungi present on ryegrass may have masked their presence and/or inhibited their growth, and that therefore, isolation of EIPFs from these sources was not possible.

The endophytic interactions of species belonging to the genera *Beauveria* and *Metarhizium* with various plants have been previously published (see the updated list of plant species by *Vega, 2018*). In this study, we isolated a strain of *B. vermiconia*, a species that was first described in *Hylamorpha elegans* by *Glare, Jackson & Cisternas (1993)*. According to our analysis, it is also able to colonize ryegrass roots. The strain *M.* aff. lepidiotae NRRL B-67994 (M25_2) is also able to colonize ryegrass roots.

The real-time PCR Miner algorithm and fluorescence microscopy indicated that native strains of *Beauveria vermiconia* NRRL B-67993 (P55_1) and *Metarhizium* aff. *lepidiotae* NRRL B-67994 (M25_2) were able to successfully colonize ryegrass roots. Both of these fungal strains can coexist endophytically inside the roots over time under various temperatures, atmospheric humidity levels and light, depending on the local climatic conditions.

Unlike most previous work (*e.g.*, *Sasan & Bidochka, 2012*; *Greenfield et al., 2016*; *Barelli, Moreira & Bidochka, 2018*; *Clifton et al., 2018*; *Cai et al., 2019*), which focused on early colonization in various plant species and EIPFs, our observations of the interaction and its impact were made over a productive cycle of ryegrass. Figures 4A and 4B shows that root colonization is season dependent. This is in agreement with the observations of *Kolczarek (2015)*, who evaluated the presence of different strains of *M. anisopliae* and *B. bassiana* in cultivated soils with *L. perenne* during autumn and spring, observing

differences between the two seasons (according to our results there was a significant increase from winter to spring coinciding with the production cycle of ryegrass), concluding that the sampling time and other factors, such as soil type and strain variety, were the determining factors.

From the beginning of our experiment, the similarity observed in the relative abundance of *B. vermiconia* P_55 and *M. aff. lepidiotae* M_25, may indicate that these strains have similar capability to colonize ryegrass roots under optimal chemical and soil moisture conditions. However, there are several factors that can affect rhizosphere and root colonization by *Beauveria* and *Metarhizium*, such as differences in nutritional availability, resulting either from fertilization treatments or differences in soil type (*Kolczarek, 2015*; *Krell et al., 2018*; *Parsa et al., 2018*; *Tall & Meyling, 2018*).

Our results suggested that soil moisture and summer conditions play an important role in the relative abundance of EIPF, although the linear mixed model indicates that the response to these abiotic factors depends on the fungus, agreeing with the results of *Borisade & Magan (2014)*. These authors also showed that *Beauveria* has lower mycelial growth than *Metarhizium* under high temperatures and different water potential levels associated with plant roots, coinciding with what was observed in the summer months in our study. Similarly, the study by *Ferus, Barta & Konôpková (2019)* recorded the higher drought tolerance of *Quercus rubra* plants subjected to dehydration and colonized by *Beauveria bassiana*, which was linked to lower stomatal conductance and relative leaf water content. The major relative abundance of *Metarhizium* compared with *Beauveria* could be explained by the specificity of this genus in colonizing plant roots (*Behie, Jones & Bidochka, 2015*), which is significant in the summer period.

There is a body of evidence indicating that the presence of EIPFs has an impact by increasing either above- and/or belowground biomass of their host plants at an early stage of development (*Jaber & Enkerli, 2016*; *Tall & Meyling, 2018*; *Ahmad et al., 2020*). This agrees with the higher fresh weight and leaf area obtained from grasslands with EIPFs in the first harvest (Fig. 4A). However, this trend was not maintained throughout the experiment. Subsequently, when we subjected the mini-pastures to water stress, their biomass production was higher than in the mini-pastures grown under optimal soil moisture conditions. As stated above, the impact of EIPFs on increasing the biomass of their host plants is well known. In this study, under our experimental conditions, it was not possible to demonstrate this effect during the last 2 months, in agreement with a previous study by *Cheplick, Perera & Koulouris (2001)* who found that drought tolerance is more dependent on ryegrass genotype than endophyte presence. However, this finding does not represent a limitation for the development of other capacities such as entomopathogenic activity.

## CONCLUSIONS

Our results indicate that native strains of *B. vermiconia* NRRL B-67993 (P55_1) and *M*. aff. *lepidiotae* NRRL B-67994 (M25_2) can colonize and co-exist on perennial ryegrass roots, and that they can maintain this association during at least one annual production cycle. In this cycle, the summer season was when we found the greatest differences between

the two strains in the colonization of ryegrass roots. It should be noted that the use of qPCR Miner proved to be an efficient method for studying the fungal colonization of ryegrass roots. In our study, although we did not observe a positive effect of EIPFs on the aboveground biomass in the ryegrass mini-pastures, we do not rule out the possibility that they could play a role in biological control of white grubs in agricultural systems, specifically in ryegrass pastures.

Therefore, the challenge for future research is to corroborate whether the presence of EIPFs in ryegrass roots can protect plants against these insect pests and thus replace the use of agrochemicals that are potentially harmful to the environment and human health.

## ACKNOWLEDGEMENTS

The authors thank Jorge Blanco for help with the mini-pastures and for providing the aboveground biomass data. We thank Luis Clasing for informing farmers where to collect plant and larva specimens. Lastly, we thank the farmers Juan Carlos Marin (Máfil) and Luis Toloza (Pichihue), and the manager Carlos Villagra (EEAA, Cabo Blanco) who allowed us to collect plant and larva specimens from their pastures. The authors are grateful to Laura Bogar and one unknown reviewer for carefully reading the manuscript and providing helpful suggestions.

### Funding

This work was supported by the Gobierno Regional Chile FIC18-55. Javier Canales was supported by the grants FONDECYT 1190812 and ANID-Millennium Science Initiative Program (ICN17-022). The funders had no role in study design, data collection and analysis, decision to publish, or preparation of the manuscript.

### Grant Disclosures

The following grant information was disclosed by the authors:
Gobierno Regional Chile: FIC18-55.
FONDECYT: 1190812.
ANID-Millennium Science Initiative Program: ICN17-022.

### Competing Interests

The authors declare that they have no competing interests.

### Author Contributions

- Milena Vera performed the experiments, analyzed the data, prepared figures and/or tables, authored or reviewed drafts of the paper, and approved the final draft.
- Sarah Zuern performed the experiments, analyzed the data, prepared figures and/or tables, authored or reviewed drafts of the paper, and approved the final draft.
- Carlos Henríquez-Valencia analyzed the data, authored or reviewed drafts of the paper, and approved the final draft.

Peerj _______________________

- Carlos Loncoman performed the experiments, authored or reviewed drafts of the paper, and approved the final draft.
- Javier Canales analyzed the data, authored or reviewed drafts of the paper, and approved the final draft.
- Frank Waller performed the experiments, analyzed the data, prepared figures and/or tables, authored or reviewed drafts of the paper, and approved the final draft.
- Esteban Basoalto performed the experiments, authored or reviewed drafts of the paper, identified scarabaeid larvae, and approved the final draft.
- Sigisfredo Garnica conceived and designed the experiments, authored or reviewed drafts of the paper, and approved the final draft.

## Field Study Permissions

The following information was supplied relating to field study approvals (*i.e.*, approving body and any reference numbers):

The farmers Juan Carlos Marin (Máfil) and Luis Toloza (Pichihue) allowed us to collect plant and insect material from their crop fields.

The manager Carlos Villagra allowed the research activity in the Santa Rosa Station Experimental of the Universidad Austral de Chile (Cabo Blanco).

## DNA Deposition

The following information was supplied regarding the deposition of DNA sequences:

MZ520361 and MZ520362

## Data Availability

The raw data is available in the Supplemental File.

## Supplemental Information

Supplemental information for this article can be found online at http://dx.doi.org/10.7717/peerj.12924#supplemental-information.

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
