# Peer review of "Exploring interactions between Beauveria and Metarhizium strains through co-inoculation and responses of perennial ryegrass in a one-year trial"

_PeerJ, doi:10.7717/peerj.12924_

## Round 0.1 · original submission · Major Revisions

I apologize for the delay, I had a difficult time finding reviewers. Your manuscript has now been thoroughly reviewed by two experts in the field and both are enthusiastic about your work. You will see that they both have some important suggestions. I believe that it should be pretty easy to address the reviewers comments and look forward to seeing a revised manuscript soon.

·

Basic reporting

In this manuscript. by Vera et al., the researchers report the results of an intriguing experiment examining the impacts of two entomopathogenic fungi (EIPFs) on the growth of perennial ryegrass, an important forage plant for pastured livestock. The writing is clear and unambiguous in nearly all places (minor exceptions noted in detailed comments below), and the literature review and context provided are sufficient. The article structure is also clear and aligned with standard scientific expectations, and the motivating questions are clearly articulated at the end of the introduction. Nearly all the raw data are provided as supplements; if the authors can add in the results of their culturing study, this manuscript will be completely aligned with the PeerJ “basic reporting” standards.

Experimental design

In this manuscript, the authors describe a thorough study on the effect of EIPFs on ryegrass, representing original, primary research that falls well within the aims and scope of PeerJ. The research questions are well defined and focus on how well the focal strains of EIPF colonize ryegrass, how their abundance changes over time, and whether their colonization of the ryegrass affects plant biomass. To my knowledge, this is a real gap in our understanding of how plants interact with entomopathogenic endophytes, and is a valuable contribution to the field.

The experiments presented represent a rigorous approach: The authors began by collecting plant and larval material from nine sites across southern Chile and cultured the fungi they contained on fungal growth media. From these, they selected strains belonging to two entomopathogenic genera (Beauveria and Metarhizium). The authors then used these strains to inoculate ryegrass plants in a “mini-pasture” experiment, which included a water restriction treatment in the summer. The authors regularly measured aboveground biomass and measured colonization by the applied entomopathogenic fungi in roots using qPCR and microscopy throughout the nine month experiment. They found that the fungi coexisted in plant roots, and that plants in the water-restricted treatment had less variable rates of fungal colonization than the plants in the wet treatment. The applied fungi did not discernibly affect plant growth in this experiment, although dry weight was slightly greater for inoculated plants than uninoculated ones in October (prior to water restriction) and in February (only for water-restricted plants). The statistical methods employed appear to be appropriate, and the methods are almost entirely described in sufficient detail for future replication. (I have requested occasional additional details in my minor comments, below.)

Validity of the findings

Although the authors have examined the effects of only two fungal strains on their ryegrass mini-pastures, their findings strike me as both valid and interesting. They have provided the underlying data for most of their analyses, although I would love to see the data for their field sampling and culturing added to the supplemental data files. I am unfamiliar with the method these authors used to calculate the relative abundance of their fungi from the qPCR data (details in "additional comments"), but a citation with more information about the method, or a shift to a more standard ratio-based metric, would address this concern. The conclusions are clearly stated and supported by the evidence presented in the paper (although the authors seem disappointed by their interesting negative result). If the authors want to dig into the connection between EIPFs and scarabaeid herbivory of ryegrass roots, it might be worth devoting some space in the discussion to outlining the most productive future experiments that could address that question directly.

Additional comments

Overall, I found this to be an interesting and well-written paper. It is easy to follow, and I think it is valuable to know that these entomopathogenic fungi can coexist in ryegrass roots as endophytes. It is also helpful to know that these endophytes did not seem to affect ryegrass growth in the mini-pastures, although, based on the framing of the paper, this may be an unexpected negative result. I appreciated the authors’ thorough approach and illustrative figures, particularly the photos of the field site, mini-pastures, and the lovely fluorescence microscopy images of endophytic fungi in roots. If the authors are able to address the questions and concerns I articulate below, I think this will be a valuable contribution to PeerJ.

Major points:
1) The authors describe an ambitious field sampling project to collect fungi for the mini-pasture experiment. However, in this version of the manuscript, they don’t report how many fungi they found overall and from how many different samples of the various substrates, what the diversity was like, or how many entomopathogenic strains they ended up getting. (They say these entomopathogenic fungi were relatively common, which is entirely possible, but I would love to see some kind of table summarizing how many they got across their sampling sites!) I would also like to know which sites and isolation material (larva, root, seed, etc.) provided the strains that they used for the experiment.
2) The negative result in this paper – that the entomopathogenic fungi did not improve plant growth – is quite interesting. Do the authors have any data on whether scarabaeid larvae were present in the mini-pastures prior to fungal inoculation, and at what densities? If the larvae were absent, it is perhaps not surprising that the fungi provided no growth benefit to the plants. I think it would be valuable to spend a little bit of time in the Discussion laying out future experiments that could help determine whether these entomopathogenic fungi can actually protect ryegrass from root herbivores like the scarabaeids.
3) The current draft of the manuscript appears to lack the required “data deposition statement” for some of the supplemental files. (It might be easiest to read if this is appended as a separate paragraph after “Conclusions,” although I believe the authors can choose to describe their data deposition strategy throughout the manuscript, as appropriate.) I appreciate the detailed information available in the supplements. The only major piece that is still missing is the results of the culturing and antagonism assays from field-collected fungi. The authors should upload those data, and also please double check that all supplemental files are mentioned and described somewhere in the manuscript, so readers can easily find them.
4) Field permits: Were Juan Carlos Marin, Luis Toloza, and Carlos Villagra the only land managers whose properties were involved in the field sampling? If so, it seems like the permissions were appropriately acquired, which is great! Please clarify this by specifying which of the nine field sites were managed by which person.


Detailed/minor comments:

In general, this manuscript reads well and tells an interesting story about a nice experiment. I noticed several typos as I read, noted below, which should be easily corrected with the spell checking feature of any commercial word processing software (e.g. Microsoft Word). The whole document should be checked, since I probably missed some minor errors.

Abstract:
L39: “… using a linear mixed-effects model and ANOVA statistical test.” This is more statistical detail than I usually expect to see in an abstract. (These would be critical details if they were new statistical approaches, but I think these are good, standard tests.) Feel free to omit if you need to tighten this section.
L41: “… isolated only in scarabaeid larvae…” Based on the methods section, I thought you isolated fungi from seeds, plant material, AND living and dead scarabaeid larvae. Did you use only strains from scarabaeid larvae, in the end? If so, please mention this in Methods.

Introduction:
L54-58: This short paragraph is a bit more broad than I find useful, as a reader, and I’m not sure what “broad yet limited” means with regard to our understanding of endophyte ecology. I’d recommend opening the paper by getting right into EIPFs as rapidly as possible. They’re so interesting, and there’s a lot we don’t know – maybe keep the first sentence that you have, just to introduce endophytes, and then go right into what is currently your second paragraph (“Some fungal endophytes…”).
L68-69: “… the origin, biology, and ecology of EIPFs are still poorly understood.” I’m sure this is true, but in this paper you can only really address a little bit of the biology and ecology of these organisms (and nothing about their origins, evolutionarily speaking). I think it would be clearer here to focus on the facets of EIPFs that you explore in this paper.
L71-79: Although this paragraph is interesting – I didn’t know anything about the evolutionary biology of EIPFs – it doesn’t seem too relevant to the rest of the paper, since you don’t try to clarify the evolutionary origins of these fungi yourselves, nor do you interpret your results in light of their evolutionary history. I think the introduction would be stronger if you focused just on the aspects of EIPF biology and ecology that you explore in the present study.
L86-87: “Thus, for example, they belong to the most common entomophatogenic fungi…” What does it mean to “belong to… fungi?” I would suggest rephrasing as follows (and fixing the typo): “These genera are among the most common entomopathogenic fungi in soils planted with perennial ryegrass,” (if, indeed, this statement is true – I’m not as familiar with ryegrass-associated fungi as the authors must be).
L101: “… where a substantial fraction of EIPF isolates was gathered.” What is a “substantial fraction,” here? The phrasing implies that you ALSO collected other EIPF isolates, but that most of them (a “substantial fraction”) came from the nine pastures mentioned earlier. If true, please mention where you got the other isolates at this point in the text. Otherwise, consider rephrasing “substantial fraction” to something else, like “many,” or, even better, naming the exact number of EIPFs you gathered.
L106: “growing in mini-pastures natural…” should read “growing in mini-pastures with natural…”
Methods:
L123-126: “The farmers….” This sentence would be a better fit in the acknowledgements section. I think you could remove it from the methods.
L136: Does “ended” mean “killed”?
L137: “Schock” should read “shock”
L144: “Isolates belonging to the genera Beauveria and Metarhizium were sequenced…” How did you identify the fungi in those genera? Please describe your morphological ID strategy, or cite a paper that describes it.
L149: “To remove fungal infections…” should read “To detect fungal infections” or “to extract fungal DNA.”
L165: “Antagonisn” should read “Antagonism”
L166: This line refers to “Table S1” as if it might contain data about the EIPF antagonism assays, but instead table S1 contains information about the bacterial and fungal load of ryegrass seeds tested! Could you please upload a table with the antagonism data?
L182: How did you choose the GTR+I+G model for this analysis? Please clarify in text.
L203: I appreciate the description of the soil used in the mini-pastures, but I would like to know more! What was the vegetation like in the area where the soil was collected? Did you expect the soil to contain scarabaeid larvae, and did you find any? Were soils sieved or otherwise homogenized before use in the mini-pastures?
L236-237: The sampling times, here called S1, S2, S3, S4, and S5, are never mentioned again in this paper. No need to tell us about these S labels if they don’t show up later!
L253-254: “… the raw threshold cycle values determine for each fungal consortium with their specific primer pairs were subtracted from those of the perennial ryegrass-specific primer pair…” This seems like an unusual way to standardize your qPCR data. Would it be possible to cite another paper that does it this way, so your readers can understand why you selected this analysis method? Alternatively, I think it may be more typical to use a ratio for this kind of relative abundance analysis, e.g. Ct_fungus divided by Ct_ryegrass, to give you a sense for the relative abundance of each fungal gene relative to the ryegrass gene.

Results:
L332: “There were significant differences…” Whenever I see this, I want to ALSO know both the magnitude and the direction of these differences. Was it a big or small difference? Which treatment had the higher values? Please be as specific as you can while reporting these results.
L337-339: “… colonization … by M. aff. lepidiotae… did not vary, whereas colonization by B. vermiconia… increased.” This is great! I’d love to see every result reported with this level of detail (contrast to line 332).
L348-349: “… from July to October…” I think it would help Northern Hemisphere readers if you could include here which season this is, e.g. “… from July to October (winter)…,” especially since you mention “the beginning of summer” at the end of the sentence.
L352: “… also a significant difference in fresh weight…” Once again: how large was the difference, and which group had a higher value?

Discussion:
L369-372: I think it would be more effective to open this section with a summary of your results and their potential implications, rather than with this general sentence about what EIPFs can do in other systems.
L378: “… EIPF communities are relatively frequent…” In the context of this sentence, it’s a little confusing to think about entire communities of EIPFs having different frequencies across a landscape. Might be clearer to say “… EIPFs are relatively common…”
L395-397: “Observations by Barelli… suggested that there is an initial period of colonization of the rhizosphere…” I don’t think I understand why you are including this detail here. Does this matter when we try to interpret your results? How might it change things? If not relevant, omit; if it is relevant, please describe a little more so your readers can follow your reasoning!
L403-404: “On the basis of these observations, we infer that both fungi are able to interact symbiotically with this forage plant…” I think you could omit this sentence, since a “symbiosis” means by definition that the two organisms grow together, which you just established in the preceding sentence. Do you mean that they interact in a commensal (or even slightly mutualistic) way? (You could make that argument, based on your growth data.) If so, please be specific.
L415: “… observing differences between the two seasons…” Which season was better, and is it consistent with what you would have expected?
L426-427: “… Beauveria was more severely affected than Metarhizium…” What direction was this effect? Did Beauveria experience a greater reduction in biomass during the dry season than Metarhizium? The discussion will be more illuminating the more specific you can be!
L431-432: I found myself getting confused here because two sentences begin with “However,” and it’s hard to keep track of the contrasts that are happening. Consider trying some different transitions to avoid multiple instances of “however” in a row.
L440-442: “… we postulate that these EIPFs could be efficient controllers of white grubs…” This seems very hypothetical, based on the results of this paper. I think it would help to be a little bit more cautious about this proposed connection: Perhaps this is a good place to say that the EIPFs, which you have demonstrated can coexist in ryegrass, might be useful in controlling grubs, and future experiments examining the impact of these strains on the grubs and ryegrass would be beneficial.

Figure 1 (legend): “(D) Ryegrass pasture effected…” should read “(D) Ryegrass pasture affected…”
Figure 3:
- How did you select the root whose images you present in this figure? Was it representative of most of the roots you looked at, or was it an especially well colonized root? Which moisture treatment was this sample from? These details could go in methods, and/or be mentioned here in the legend.
- What are the units on the y axis? Is this… the ratio of (Ct_fungus/Ct_ryegrass)? Or maybe (Ct_fungus – Ct_ryegrass)? Please label specifically.

Supplements:
- I can’t find a mention of Supplemental Figure 1 in the main text. Could you please mention it, where appropriate, so interested readers know to look for it?
- All supplemental tables: It would be helpful to have some kind of header or brief description of the table’s contents in each supplemental file.
- Table S2 may have a typographic error: pH (in water) is listed as 60. Should this read 6,0?
- Table S3: What does the asterisk mean after the header “Larvae”? Please clarify in the legend.
- Table S4: Column D header should probably read “Efficiency,” rather than “Efficience.”

Reviewer 2 ·

Basic reporting

Line 101: Change “was” to “were” because the word “isolates” is plural.

Line 103: This study did not explore how EIPF strains can act as a biological control of white grubs. Please remove the last phrase.

Line 137: Change “schock” to “shock”.

Line 163: There is a missing period.

Line 165: “Antagonism” is spelled incorrectly.

Line 166: The first two columns in Table S1 are confusing. What do the codes under “Seeds” and “Endophyte” mean? Please remember that readers are not familiar with the codes you assigned during your experimental design. Please re-name or reformat this table to be more intuitive. Perhaps you could list locations that correspond to the map in Figure 1 where you collected the seeds and fungi.

Line 298: Please provide more information on the linear mixed model. What were the fixed effects? What were the random effects? How did you choose among statistical models (e.g., AIC)? How did you check that the assumptions of the model were met (e.g., residuals normally distributed, etc.)?

Line 201-203: Readers not based at your university do not know what “local climatic conditions present in the EEAA belonging to the Universidad Austral de Chile located in sector Cabo Blanco” are. Please report the actual values for precipitation, temperature, etc.

Line 326: It is not clear from Tabe S4 which samples were inoculated vs. uninoculated, especially for readers that are not familiar with reading qPCR values. Could the authors add an extra column to clarify?

Comments related to Figure 3: There is no Figure 3C. It skips from 3B to 3D. Please re-format. What is the y-axis labeled “Relative Colonisation Rate”? Is this different from “Relative Abundance” in Table S4? Rate implies a unit per unit time, but the x-axis is categorical for a given month. How can you have a “rate” from one qPCR abundance reading? Was it based on the change from the previous month? Why aren’t there statistical codes above each bar - why are some blank?

Figure 3A-B: “Different letters indicate significant differences among treatment groups”. I’m confused. What is the treatment group for 3A? Are you referring to month or fungal taxon? For 3B, I’m confused because the term “treatment group” seems to imply 80-85% FC vs. 40-45% FC but the statistical codes don’t seem to match drought treatment. For example in 3B, “c” is the same for fungal taxa WITHIN but not among drought treatments whereas “a” vs. “b” seems to signify differences between the abundance of two fungal taxa WITHIN but not among a drought treatment. Please clarify what the statistical codes are referring to.

Line 332: There is a missing comma between “June” and “August”.

Figure 4: Again, I’m confused why there aren’t statistical codes above each bar. Some are blank.

Line 350: Is the Tukey test a pairwise comparison between July and October? You mention a decrease towards the beginning of summer at the end of the sentence, so it’s not clear which two months you’re comparing.

Line 350: Usage of “however” means that you’re contradicting the preceding sentence. Differences in dry weight in October do not contradict fresh weight findings in the previous sentence. These are two different findings that are not directly comparable.

Line 355: “was not significantly so under optimal soil moisture conditions” Could the authors rephrase this? I’m not sure what “not significantly so” means.

Line 364: Replace the comma with a period.

Line 365: What are those three p-values comparing? Is it control vs. inoculated per month? If so, why are there only three p-values? I’m confused why Figure 4E is cited… there is no water stress treatment in 4E.

Lines 389-392: These sentences are not particularly helpful for a reader. Please explain what mechanisms of root colonization or plant-endophyte interactions are relevant to the findings you discuss in Lines 392-399. How do these cited works contextualize your findings?

Line 432: Avoid using “however” as a transition for successive sentences.

Experimental design

Line 150: I’m a bit confused about the sequencing of the ryegrass seeds. The authors extracted total genomic DNA from seeds and used fungal-specific primers to amplify the ITS region with PCR. The “automated genetic analyzer” sounds like Sanger sequencing technology, unless I am mistaken? How can you Sanger sequence when amplifying from total genomic DNA? Couldn’t there be multiple fungi living within the seeds?

There’s no rationale for why these plants were co-inoculated rather than separately inoculated with each endophytic strain. I understand that the qPCR method allowed the authors to quantify the abundance of each fungal strain but it seems odd to not have treatments where plants were inoculated with just one strain. The authors state that they were unable to isolate these fungal genera from ryegrass roots and seeds, so I’m wondering why they assumed that fungal co-occurrence on plant roots was more important to explore than separate inoculation.

Validity of the findings

The two main response variables in this study, fungal abundance and aboveground biomass, were quantified with sound methods.

Additional comments

I appreciate the work undertaken by these authors. This field experiment tested how fungal abundance and drought stress affected plant growth. The results showed interesting temporal patterns in fungal abundance for two understudied fungal endophyte species and how changes in fungal abundance correlated with plant growth traits that further our understanding of the ecology of these fungal taxa. However, I have numerous questions and concerns that should be addressed before this article can be published that I discuss in the various sections of this review.

The effect of drought on ryegrass is introduced rather abruptly at the end of the introduction. What is it about this system that makes understanding drought important? For example, are plants typically drought stressed during certain seasons (e.g., summer) and did that inform the experimental design? Do these plants have adaptations for drought stress? Is there evidence in the literature that drought stress has impacted the population dynamics for this plant species or that EIPF can mitigate drought stress for plants? While the importance of drought to ryegrass may be obvious for the authors, it is not obvious for the readers. Please explain and build a case for why your study explored drought in the context of EIPF interactions.

Why were there no measurements of belowground plant biomass? EIPF likely have more direct effects on roots than shoots.

I enjoyed and appreciated the microscopy images the authors provided of endophyte root colonization.

---

## Round 0.2 · Minor Revisions

Thank you for your revisions. Can you please make the minor changes suggested by the reviewer and do a final careful proof reading for typographical errors?

·

Basic reporting

The revised manuscript is greatly improved, and satisfactorily addressed my concerns from the prior review. I thank the authors for their detailed responses to my comments. The updated manuscript is much clearer, and will be an interesting publication. There are still occasional typographic errors, but I suspect those will be easy to take care of in the final formatting stages. I only have a couple of additional minor comments which I think will improve the manuscript.

L23: “most of such pastures” should be changed to “most such pastures.”
L138: I am still kind of confused by the use of the word “remove” here. Does this mean that you sterilized the seeds before grinding, in order to remove all fungal infections throughout the seed? (But -- you wanted to preserve endophytic infections, right?) Or were you sterilizing just to remove fungi on the outsides of the seeds? I think this would be clear by adding a comma: “To remove external fungal infections on ryegrass seeds, 1g of seeds were surface sterilized as described above, and then ground under liquid nitrogen as with the mycelium.” Or something like that? If I still have not managed to capture your meaning here, I would request some kind of rephrasing, if possible, just so it’s a little clearer for future readers.
Table S1: Thank you for adding these data to your submission! Would it be possible to add one more column to this table, indicating which of these strains you identified as EIPFs? This is not strictly required, but I suspect some readers may find it helpful.
Table S3: Please add column headers to this table. Additionally, the first pH entry still reads “60.” Please change to the correct value (likely 6,0).

Experimental design

As before, the experimental design seems appropriate to me, and relevant to the Aims and Scope of PeerJ. This is an interesting study that addresses a relevant knowledge gap.

Validity of the findings

The findings presented here appear to be valid and the conclusions seem justified based on the data.

Additional comments

The authors have done a very nice job revising this paper. Every section has been improved since the last version -- I commend their efforts! I think this paper will be a useful reference for future researchers.

---

## Round 0.3 · accepted · Accept

Thank you for addressing the minor revisions. I am recommending your manuscript for publication.